# The CINs of Polo-Like Kinase 1 in Cancer

**DOI:** 10.3390/cancers12102953

**Published:** 2020-10-13

**Authors:** Chelsea E. Cunningham, Mackenzie J. MacAuley, Frederick S. Vizeacoumar, Omar Abuhussein, Andrew Freywald, Franco J. Vizeacoumar

**Affiliations:** 1Department of Pathology, College of Medicine, University of Saskatchewan, Saskatoon, SK S7N 5E5, Canada; mjm341@mail.usask.ca (M.J.M.); frederick.vizeacoumar@usask.ca (F.S.V.); 2College of Pharmacy, University of Saskatchewan, 104 Clinic Place, Saskatoon, SK S7N 2Z4, Canada; oea769@mail.usask.ca; 3Cancer Research, Saskatchewan Cancer Agency, 107 Wiggins Road, Saskatoon, SK S7N 5E5, Canada

**Keywords:** Polo-like kinase 1, chromosomal instability, DNA damage repair, synthetic dosage lethality

## Abstract

**Simple Summary:**

Many alterations specific to cancer cells have been investigated as targets for targeted therapies. Chromosomal instability is a characteristic of nearly all cancers that can limit response to targeted therapies by ensuring the tumor population is not genetically homogenous. Polo-like Kinase 1 (PLK1) is often up regulated in cancers and it regulates chromosomal instability extensively. PLK1 has been the subject of much pre-clinical and clinical studies, but thus far, PLK1 inhibitors have not shown significant improvement in cancer patients. We discuss the numerous roles and interactions of PLK1 in regulating chromosomal instability, and how these may provide an avenue for identifying targets for targeted therapies. As selective inhibitors of PLK1 showed limited clinical success, we also highlight how genetic interactions of PLK1 may be exploited to tackle these challenges.

**Abstract:**

Polo-like kinase 1 (PLK1) is overexpressed near ubiquitously across all cancer types and dysregulation of this enzyme is closely tied to increased chromosomal instability and tumor heterogeneity. PLK1 is a mitotic kinase with a critical role in maintaining chromosomal integrity through its function in processes ranging from the mitotic checkpoint, centrosome biogenesis, bipolar spindle formation, chromosome segregation, DNA replication licensing, DNA damage repair, and cytokinesis. The relation between dysregulated PLK1 and chromosomal instability (CIN) makes it an attractive target for cancer therapy. However, clinical trials with PLK1 inhibitors as cancer drugs have generally displayed poor responses or adverse side-effects. This is in part because targeting CIN regulators, including PLK1, can elevate CIN to lethal levels in normal cells, affecting normal physiology. Nevertheless, aiming at related genetic interactions, such as synthetic dosage lethal (SDL) interactions of PLK1 instead of PLK1 itself, can help to avoid the detrimental side effects associated with increased levels of CIN. Since PLK1 overexpression contributes to tumor heterogeneity, targeting SDL interactions may also provide an effective strategy to suppressing this malignant phenotype in a personalized fashion.

## 1. Introduction

Tumor heterogeneity and an increased rate of genetic mutations are prevalent features of cancer that need to be addressed in therapy to prevent treatment resistance and improve patient outcomes. Tumor heterogeneity refers to the presence of multiple distinct genotypes and phenotypes within the tumor cell population. Heterogeneity can exist within a single tumor, referred to as intra-tumor heterogeneity, or between patients of the same cancer type, defined as inter-tumor heterogeneity [1]. It contributes to drug resistance, tumor relapse and overall worse prognosis [1,2], and presents a hurdle for successful application of precision medicine and targeted therapies. One of the key driving force of tumor heterogeneity is chromosomal instability (CIN), a form of genomic instability that is frequent in many cancers, affecting up to 80% of solid tumors [3,4]. Addressing the source of the plasticity and resistance itself, such as targeting CIN, may prove to be a more efficient approach over targeting specific alterations, individually and in combination, as they arise. Targeting CIN in this way may allow to suppress tumor evolution, which usually complicates treatment and increases heterogeneity.

Chromosomal integrity is monitored carefully by cells throughout the cell cycle and there are multiple checkpoints dedicated to ensuring mitosis results in a faithful distribution of the genetic material to daughter cells. Normally, these checkpoints ensure cell division results in two healthy daughter cells with the complete intact genetic material, however in cancer, the checkpoints can become compromised, resulting in genetic defects of the daughter cells. Checkpoint defects promote mishaps in DNA replication and cell division, leading to CIN and aneuploidy. Processes coordinated throughout the cell cycle that, when dysregulated, can contribute to CIN include: double strand DNA break repair [5], DNA replication fork resolution [6], kinetochore-microtubule (MT) attachment formation [7], centrosome maturation and positioning [8], bipolar spindle formation [9], sister chromatid alignment and segregation [10], cleavage furrow formation [11], and telomere maintenance [12].

The key players regulating the cell cycle, and therefore CIN, are the multiple mitotic kinases within the cell. These kinases drive regulatory feedback and signaling loops that either arrest the cell cycle, or drive progression through the cell cycle checkpoints. Additionally, these kinases are often act as parts of regulatory feedback loops downstream of the cell cycle checkpoints to ensure commitment to cell cycle progression once checkpoint requirements have been cleared. These kinases include: cyclin dependent kinase 1 (CDK1 or CDC2), polo-like kinase 1 (PLK1), the Aurora kinases A, B, and C (AURKA/B/C), NIMA related kinase 2 (NEK2), Bub1, BubR1 (or Bub1B), and TTK (or MPS1) [13]. Wee1 and Myt1 (or PKMYT1) kinases oppose the phosphatase activity of cell division cycle 25C phosphatase (CDC25C) to activate Cyclin/CDK1 complexes, but are not considered mitotic kinases themselves, as they play a signal transduction role [13]. PLK1 and CDK1 with Cyclin B are required for entering into mitosis [14]. NEK2 activity drives centrosome dissociation, initiating the centrosome cycle [15]. AURKA, AURKB, and PLK1 are important for regulating spindle dynamics and chromosome attachments [16], with AURKA activity also regulating central spindle microtubule dynamics [17]. AURKB also regulates centrosome clustering for bipolar spindle formation [18]. TTK and BubR1 are critical components of the spindle assembly checkpoint (SAC) [19]. CDK1 and PLK1 work at the SAC to control anaphase promoting cyclosome proteasome complex (APC/C) activation [20,21]. Lastly, PLK1 and AURKB signaling initiates cytokinesis [22]. Working in concert, these kinases regulate a multitude of cell cycle checkpoints, effector proteins, and each other and their deregulation is associated with multiple malignancies [13,23]. Many of these molecules have been found to be overexpressed in tumor cells and have been put forward as potential therapeutic targets, including CDK1 [24], NEK2 [25], AURKA [26], Bub1 [27], and TTK [28]. In fact, our own analyses of the TCGA data across multiple cancers show their abnormal expression in various tumor types (Figure 1A). A precise understanding of their role in malignancy should provide novel opportunities to exploit them for treatment purposes without affecting normal cells that have intact cell cycle checkpoints. PLK1 in particular, is a key player that has been well-characterized and with further study may provide valuable options for developing novel targeted therapies. The extensive crosstalk between PLK1 and the Aurora kinases has been well studied and PLK1 carries out many of its functions in cooperation with the Aurora kinases, but the complexities of this relationship is reviewed elsewhere, and the functions of PLK1 will be the focus of this text [29]. The processing and timing of key cell cycle events can become dysregulated, leading to CIN through abnormal PLK1 activities, and in this review, we discuss therapeutic avenues that may arise from near-ubiquitous PLK1 deregulation in cancer cells.

## 2. Regulation and Activity of PLK1 in the Cell Cycle

PLK1 is a serine/threonine (Ser/Thr) protein kinase belonging to the PLK family, which overall consists of 5 kinases, PLK1 to PLK5 [29]. The structure of PLK1 includes the kinase domain, and two polo-box motifs composing the polo-box domain (PBD) (Figure 1B). The PBD of PLK1 is the hallmark of the PLK family. This PBD primarily recognizes and binds substrates that contain a consensus PLK1 phosphorylation motif, although regulation of some PLK1 partners do not seem to be reliant on their phosphorylation [30,31,32]. Many molecules interacting with PLK1 PBD are controlled through priming phosphorylation by cyclin B/CDK1 to generate the consensus PDB-binding phospho-motif, which allows PLK1 and cyclin B/CDK1 to cooperatively regulate the cell cycle [20,33,34,35].

PLK1 is primarily expressed in actively dividing cells and is barely detectable in non-proliferating cell types, which is consistent with its role in cell cycle progression [36,37,38]. PLK1-null mice are embryonic lethal, with embryos arrested after the eight-cell stage [39], suggesting a critical role for this kinase in early dividing cells and embryo development. PLK1 expression is regulated at the mRNA level by a cell cycle gene homology region in the promoter that represses PLK1 mRNA production during the G1 phase [40]. PLK1 protein levels peak in G2 and remains high until cytokinesis and mitotic exit [41,42]. PLK1 protein degradation is mediated through the destruction box (D-box) motif and subsequent proteasome targeting in G1/S to keep PLK1 levels low [43]. Heat shock protein 90 (Hsp90) helps stabilize and accumulate PLK1 protein prior to mitotic entry [44]. PLK1 is also regulated at the post-translational level, with the activating loop within the PLK1 kinase domain only being phosphorylated upon entry to mitosis [41,45]. This phosphorylation event releases PLK1 from an autoinhibitory interaction between its catalytic and PBD domains, and has been shown to be carried out, at least in some cases, by BORA and AURKA kinases [30,42,46]. Interestingly, phosphorylation at two sites in the activation loop, S137 and T210, increases PLK1 kinase activity with different timing kinetics [47]. While either phosphorylation event triggers partial PLK1 activation, phosphorylation at both sites synergistically increases its catalytic action [47].

In addition to the modulation of its kinase activity, PLK1 localization changes throughout the cell cycle, moving from the cytosol to the centrosome and spindle poles in early mitosis to kinetochores in metaphase cells to central spindle at anaphase and midbody in cytokinesis [41,48,49]. This dynamic localization pattern is partially directed by interactions of its PBD with cyclin B/CDK1-primed phospho-substrates. PLK1 also localizes to the nucleus prior to entering mitosis through its nuclear localization signal (NLS). Disrupting the PLK1 NLS causes cells to arrest in the G2 phase [50]. It is interesting to note that kinase-defective PLK1 stably associates with centrosomes and this centrosome associated PLK1 prevents mitotic re-entry of cells with DNA damage [51]. Thus, it appears that PLK1 drives cells through distinct stages of the cell cycle by acting within different cellular compartments. Below, we discuss how PLK1 associates with different cellular compartments to regulate normal cell cycle progression without compromising the integrity of the genome.

## 3. The Role of PLK1 in Driving Cell Cycle Progression

### 3.1. PLK1 and DNA Replication

Premature cell cycle progression can drive replication stress and lead to double strand DNA breaks at unresolved single strand DNA replication forks. This can be driven by overactive mitotic kinases, including PLK1 [52]. To prevent this replication stress, the S-phase ATR checkpoint suppresses forkhead box protein M1 (FoxM1) until the G2 phase to inhibit mitotic entry and control expression of PLK1 and Cyclin B [53]. Accumulation and activation of PLK1 begins at the completion of S-phase [54]. Recent evidence shows that PLK1 is suppressed by active DNA replication and the completion of DNA replication releases this suppression, leading to increased activation of PLK1 through phosphorylation and triggering mitotic progression. This notion is supported by the finding that inhibitors of CHEK1, the key ATR effector, uncouples the sequential activation of PLK1 until after DNA replication is complete [52].

PLK1 can promote DNA replication in some contexts such as DNA damage-induced stress. Origin recognition complex subunit 2 (ORC2) phosphorylation by PLK1 helps promote DNA replication initiation and formation of the pre-replicative complex under stress, such as DNA-damaging ultraviolet (UV) irradiation [55]. PLK1 may also promote DNA replication through regulation of the DNA replication inhibitor geminin, as depletion of PLK1 was shown to slow DNA synthesis and stabilize geminin [56]. This suggests PLK1 can promote the progression of DNA replication, potentially even in the presence of replication stress.

### 3.2. PLK1 and Mitotic Entry

Cyclins and their CDKs are the drivers of the cell cycle, moving the cells forward towards successful cell division through signaling the transition through checkpoints via positive feedback loops. Cyclin B/CDK1 is the main kinase for mitotic entry following satisfaction of the G2/M phase checkpoint along with PLK1. PLK1 phosphorylates cyclin B in its nuclear export sequence, sequestering it in the nucleus prior to mitotic entry [57] and also phosphorylates CDC25C, which in turn activates cyclin B [58]. PLK1 and Cyclin B/CDK1 cooperatively act to positively regulate FoxM1. FoxM1 is a transcriptional regulator of G2/M phase genes. FoxM1 expression begins to increase in S phase and is phosphorylated by Cyclin B/CDK1 and PLK1 to increase its transcriptional activity of mitotic targets, such as AURKB, cyclin B, and PLK1 itself, in a positive feedback loop in preparation to irreversibly commit cells to mitotic entry [33].

### 3.3. PLK1 and Mitotic Entry Following DNA Damage

PLK1 is also a main component of the mitotic re-entry mechanism following cell cycle arrest. After prolonged DNA damage and subsequent cell cycle arrest, cells can adapt by silencing the checkpoint signaling to continue dividing despite the continued presence of DNA damage. This is referred to as checkpoint adaptation often found in tumor cells and results in increased genomic instability [59]. Phosphorylation by active PLK1 helps drive G2 checkpoint recovery by attenuating the ATM/ATR response and allowing mitotic entry despite DNA damage [60]. PLK1 acts on the DNA damage sensing Mre11/Rad50/Nibrin (MRN) complex, upstream of the ATM/ATR DNA damage checkpoint. Phosphorylation of the Mre11 subunit of the complex by PLK1 blocks activation of the checkpoint, or stops checkpoint signaling to allow re-entry into the cell cycle [61]. PLK1 kinase activity also drives G2 checkpoint adaptation following DNA damage by promoting the movement of p53 out of the nucleus and by promoting p53 degradation through ubiquitination [62,63]. PLK1 is also able to inhibit p53 transcription [64]. Thus, the increasing activity of PLK1 seems to be a mechanism by which cancer cells can continue cell division despite DNA damage, resulting in increased levels of CIN.

## 4. Role of PLK1 during Mitosis and Cytokinesis

### 4.1. PLK1 and Centrosome Function

Accurate chromosome segregation requires coordination of centrosome positioning, chromosome condensation, nuclear envelope breakdown, kinetochore-MT attachment, and spindle tension. In the normal centrosome cycle, centrioles disengage and duplicate to form two separate centrosomes, the centrosomes undergo elongation and maturation where they recruit components required for MT nucleation, and finally, separately migrate to opposite poles to form the bipolar spindle required for chromosome segregation in coordination with the cell cycle. In this context, CIN can arise from two mechanistically distinct scenarios: centrosome amplification, meaning the centrioles duplicate more than one time, or errors in centrosome positioning. Increased PLK1 expression is correlated with the presence of multiple centrosomes, suggesting it regulates centriole duplication early on in the centrosome cycle [2,65]. PLK1 and Separase activities are both required for cleavage of cohesin between centrioles, resulting in disengagement of centrioles in anaphase to allow for centriole duplication in the following interphase [66]. Centrosome duplication occurs once per cell cycle, so increased expression and activity of PLK1 could perceivably allow this process to occur inordinately. And indeed, PLK1 activity has been shown to drive centriole disengagement, allowing centriole reduplication outside of the typical centriole duplication [67]. Constitutively active PLK1 has been shown to protect against DNA damage-induced centrosome amplification in BRCA1-deficient cells [68]. Additionally, loss of Liver Kinase B1 (or STK11) leads to centrosome amplification through increased phosphorylation and activation of PLK1 [69], and BubR1 expression prevents centrosome amplification through inhibition of PLK1 [70]. The specific role of PLK1 in centrosome amplification requires more investigation, but PLK1 seems to be an important player in regulating centrosome number.

In addition to centrosome duplication, PLK1 also regulates centrosome maturation through recruitment of the pericentriolar material including the γ-tubulin ring complex (γ-TuRC). The γ-TuRC allows for MT nucleation, which forms the actual spindle fibers. The centrosomes and newly forming spindles separate to form the bipolar spindle essential for proper chromosome segregation. PLK1 and a PP1 subunit, MYPT1, work antagonistically to recruit γ-tubulin to the centrosomes required for MT nucleation, with PLK1 activity promoting recruitment [71,72]. Another way PLK1 promotes centrosome maturation for bipolar spindle formation is through the displacement of ninein-like protein (Nlp or NINL), an important component of the centrosome which promotes non-mitotic spindle MT nucleation during interphase by recruiting γ-TuRC, in later cell cycle [73]. PLK1 additionally regulates recruitment of γ-TuRC to the centrosome through the γ-TuRC targeting factor, NEDD1, by direct phosphorylation and indirectly through HAUS augmin-like complex subunit 6 (HAUS6 or FAM29A), which regulates NEDD1 localization [74,75]. PLK1 also interacts with and helps localize the MT-associated protein 9 (MAP9 or ASAP) to the centrosome, which again has a role in recruiting γ-TuRC [76]. PLK1 further stabilizes the centrosome structure for spindle formation through phosphorylation of the centrosome protein, Kizuna, and loss of Kizuna results in fragmented pericentriolar material and diffuse defective spindles [77]. All functions together highlight the importance of PLK1 in coordinating the number and formation of correct spindles originating from the centrosomes for equal bipolar cell division.

Centrosome clustering is a process that prompts centrosome-amplified cancer cells to form a pseudo-bipolar spindle that is mitosis competent, as multipolar cell divisions are often inviable [8]. Although chromosomal passenger complex and kinetochore tension sensing components have been proposed to be essential for centrosome clustering [18], PLK1 along with Stat3 and Stathmin, has also been directly linked to centrosome clustering by regulating γ-tubulin levels [78] as well as in the recruitment of Eg5 motor proteins [79,80], to promote cells through mitosis. Disruption of this process in particular is expected to have potent anticancer activity [81], and PLK1 may provide a mechanism to target centrosome clustering.

### 4.2. PLK1 and Chromosome Alignment

Sister chromatid cohesion and condensation are essential for proper diploid daughter cell formation. This is regulated by condensin I and II complexes, which both contain structural maintenance of chromosomes (SMC) proteins that form a ring structure and bind linear chromatin. Condensin I is present in interphase, while condensin II is active in mitotic cells. PLK1 regulates the levels of the CAP-H2 component in condensin II in early mitosis by phosphorylating CAP-H2 and protects condensin II from APC/C degradation [82]. Dysregulation of condensin II through PLK1 inhibition leads to anaphase segregation defects such as lagging chromosomes [82]. PLK1 also binds to condensin II through CAP-D3, which is primed by the initial CDK1-dependent phosphorylation, to promote further phosphorylation and condensin II activity [83]. PLK1 also phosphorylates the histone kinase Haspin, which in turn generates phospho-histone marks, like phosphoH3T3 to induce activation of AURKB and other chromosomal passenger proteins, which guide chromosome structure in mitosis [84,85].

Apart from regulating the condensin complex, PLK1 also regulates cohesin on the chromosome arms by phosphorylating this protein to promote its dissociation and thereby, dictates chromosome separation [86,87]. PLK1 also works with the PLK1-interacting checkpoint helicase (PICH or ERCC6L) to colocalize at the kinetochores with cohesion and loss of PICH leads to chromosome disorganization [88]. PLK1 acts to drive chromosome segregation forward, and one of the main functions of the SAC is to counteract PLK1-driven cohesin dissociation through PP2A antagonistic activity until all requirements for correct chromosome segregation are satisfied [87,89].

Paradoxically, while the SAC inhibits PLK1 activity, PLK1 has a role in establishing the SAC complex. PLK1 works with the dual specificity TTK protein kinase to recruit SAC components, Mad1 and Mad2. Dual inhibition of PLK1 and TTK leads to mitotic slippage or premature cell cycle progression, indicative of a weakened checkpoint response [90]. Normally, TTK signals to activate the SAC in response to unattached kinetochores, which signals Bub1 to recruit other SAC components, including PP2A, shugoshin 1 (SGO1) and interestingly, also PLK1, to the centromere [21,89]. PLK1 also acts upstream of TTK, where it needed for the full activation of this kinase [21]. PLK1 recruited by Bub1 phosphorylates Met-Glu-Leu-Thr (MELT) repeats of a kinetochore scaffold protein Knl1 [21], and these phospho-motifs act as scaffolding to recruit additional SAC components [91]. Defects in SAC function result in defective chromosome segregation, and this is another mechanism through which PLK1 dysregulation contributes to CIN.

### 4.3. PLK1 and Kinetochore-Microtubule Dynamics

In addition to its role in chromosome alignment, PLK1 has also been implicated in the maintenance of Kinetochore-MT attachments. The cytoplasmic linker protein CLIP-170 recruits PLK1 to kinetochores and loss of this interaction promotes kinetochore attachment defects [92]. PLK1 also recruits the MIS12 kinetochore complex assembly cochaperone protein, SUGT1 (or Sgt1) to kinetochores, which initiates the process of forming the kinetochore-MT attachments [93]. PLK1 activity at the kinetochores directly decreases MT dynamics, thereby stabilizing the kinetochore-MT attachments [94], and overly strong stabilization of MTs by overactive PLK1, promotes mis-attachments leading to CIN [95]. However, too dynamic or lax an attachment is also detrimental to cells if it results in insufficient tension to evenly pull apart sister chromatids. MYPT1 destabilizes kinetochore-MT attachments and this action is, at least in part, through dephosphorylation of PLK1 at kinetochores [96]. Cyclin A/CDK1 phosphorylates and recruits phosphatase MYPT1, a PP1 complex subunit, to antagonize and dephosphorylate PLK1, and increase the dynamic nature of the kinetochore-MT attachments [96]. Dysregulation of PLK1 can also contribute to CIN by prematurely generating kinetochore-MT attachments, leading to errors in chromosome segregation. Therefore, it can be concluded that dysregulation of PLK1 in either direction is sufficient to drive defects in the kinetochore attachments, thereby increasing chromosome segregation defects ultimately resulting in CIN.

Phosporylation of the kinetochore components is maintained by SAC signaling until sufficient spindle tension is present in the kinetochore-MT attachments for successful chromosome segregation at which time, the phosphorylation is removed and the signaling cascade activates APC/C [97]. PLK1 knockdown cells are still able to form kinetochore-MT attachments [98], however, PLK1 is required for maintenance of the inhibitory kinetochore tension-sensing phosphorylation epitope known as 3F3/2, which signals inhibition of APC/C by the SAC [97,99]. PLK1 also phosphorylates nuclear distribution protein C (NudC), causing its accumulation in early mitosis or when tension at the kinetochores is lost, which helps to maintain chromosome alignment until tension is achieved [100]. Interestingly, phosphorylation of PLK1 at serine 137 (S137) has been shown to bypass the constraints of the SAC [47,94]. This would further promote CIN in the context of PLK1 overexpression.

### 4.4. PLK1 and Cytokinesis

Just after the chromosomes successfully segregate during anaphase, interpolar spindles begin to accumulate and overlap with opposing directionality in the equatorial plane between the two spindle poles to form the central spindle. The central spindles are able to push against one another through the action of MT-associated proteins, such as PRC1 and centrosome protein CEP55, and kinesin motor proteins, such as kinesin family member 20A (KIF20A or MKlp2) [101]. PLK1 localizes to the central spindle with KIF20A following metaphase [102], but is not strictly required for central spindle formation [49]. PLK1 also interacts with other kinesin family members, such as KIF2C (or MCAK) and KIF23 (or MKlp1), which are important for the movement of the spindles between daughter cells and maintaining chromosome stability [103,104,105]. Prolonged PLK1 activity caused by blocking either dephosphorylation or degradation delays mitotic exit [106], while overexpression of PLK1 leads to abscission defects [107]. PLK1 inhibition can lead to cell division without cytokinesis through mis-localization of anillin and myosin II and the small GTPase Ras homolog gene family, member A (RhoA), which leads to contractile ring defects and consequently, binucleate cells [49]. This suggests PLK1 helps to ready the cell for the formation of the contractile ring and correct abscission.

The timing and location of the contractile ring and cleavage furrow ingression is critical for even distribution of the daughter cell chromosomes and organelles. Cytokinesis in cells with incompletely separated chromosomes can result in aneuploid daughter cells, as seen with PLK1 dysregulation through inhibition of PLK1 activity and through PLK1 overexpression, both resulting in binucleate and polyploid cells [49,107]. Relevant to this, PLK1 regulates the timing and location of RhoA signaling to form the contractile ring and initiate cleavage furrow ingression driven by the MT motor complex centralspindlin [108]. PLK1 regulates recruitment of centralspindlin through PRC1 [108]. PLK1-produced phosphorylation forms the docking site in the central spindle for epithelial cell transforming 2 (Ect2), a Rho-specific guanine nucleotide exchange factor important for cleavage furrow formation [109,110]. Correct localization of Ect2 to the central spindle is required for correct positioning of the RhoA signaling cascade and the resulting cleavage furrow ingression [49]. PLK1 also stabilizes CEP55 and causes it to dissociate from the centrosome, allowing it to re-localize to the midbody just prior to abscission [111]. These movements are tightly regulated to prevent an early initiation of cytokinesis [112,113]. Dysregulation through PLK1 overexpression leads to mis-localization of CEP55 and disrupts abscission, also resulting in binucleate cells [107]. Timing of cytokinesis until after chromosome segregation is important to prevent CIN and aneuploidy and is partially regulated by PLK1.

## 5. PLK1 and Maintenance of DNA Integrity

### 5.1. PLK1 and DNA Damage Checkpoint Function

Before cells enter mitosis, the DNA damage checkpoint monitors any injury to DNA molecules to prevent mitotic entry, while DNA damage is present. ATR, ATM, and DNA-dependent protein kinase catalytic subunit (DNA-PKcs) are the main mediators that activate the signaling cascade to arrest cells in the presence of DNA damage and work in concert with the mitotic kinases, including PLK1. Inhibiting ATM-induced arrest leads to increased PLK1 kinase activity, suggesting ATM inhibits PLK1 [114,115]. Conversely, ATM activity in response to DNA damage inhibits activation of PLK1 by phosphorylation and promotes PLK1 dephosphorylation by PP2A [116,117]. ATR-mediated checkpoint response has not been shown to induce the same effect in all cases [116], but ATR has been shown to be capable of inhibiting PLK1 in ATM-deficient cells [115]. Another cell cycle kinase, AURKA, and its activator, BORA, phosphorylate PLK1 at T210 to activate PLK1, and this phosphorylation is inhibited in response to DNA damage [118]. Increased PLK1 activity opposes checkpoint signaling and is required for re-entry into mitosis following DNA damage induced G2/M phase arrest resulting from UV irradiation [119]. DNA-PKcs physically interacts with PLK1 and promotes its activation through phosphorylation. PLK1 also phosphorylates DNA-PKcs in return, in preparation for mitotic entry [120,121]. Additionally, PLK1 opposes activation of a cell cycle arrest effector, the CHEK2 kinase, which acts downstream of ATM and DNA-PKcs [119]. One of CHEK1 upstream regulator proteins, Claspin, which is an adaptor protein that binds both BRCA and DNA, is also negatively regulated by PLK1 [122]. Mitotic re-entry following DNA damage and inhibition of ATM can also be driven by increased phosphorylation of PLK1 by AURKA [123]. These checkpoints function to control cell cycle arrest through regulating PLK1 activity in response to DNA damage, thereby preventing mitotic progression. PLK1 also interacts directly with sensors of DNA damage to regulate cell cycle arrest. PLK1 phosphorylates the Rad9 component of the Rad9A-Hus1-Rad1 (9-1-1) protein complex that localizes to sites of DNA damage and promotes checkpoint-mediated arrest through ATR and double strand DNA break (DSB) repair mechanisms. The phosphorylation of Rad9 by PLK1 suppresses this checkpoint activation [124]. DNA damage during mitosis has also been found to inhibit PLK1, although this mechanism was found to be both ATM and p53 independent [125].

### 5.2. PLK1 and DNA Damage Repair Pathways

PLK1 is a modulator of the homologous recombination (HR) pathway, acting on both the proteins that comprise the MRN complex and the Rad51 recombinase. Within the MRN complex, PLK1 phosphorylates Mre11 at residue S649, which can inhibit MRN complex localization and recruitment of both HR and non-homologous end-joining (NHEJ) repair proteins to sites of DNA damage [61]. This allows for premature checkpoint termination, and consequently re-entry into the cell cycle despite unrepaired lesions [61]. Conversely, PLK1 directly phosphorylates the Rad51 recombinase, stimulating a cascade of events that facilitate further phosphorylation of Rad51 by casein kinase 2 (CK2) and resulting in binding to the MRN complex componentNbs1 [126]. This response increases the presence of Rad51 at sites of DNA damage and enhances the overall responsiveness of the HR pathway, contributing to cell proliferation and tumorigenesis. PLK1 also phosphorylates BRCA2, a stabilizer of Rad51 that promotes HR, and this phosphorylation is inhibited by DNA damage to promote HR-mediated repair [127]. Aberrant PLK1 phosphorylation of BRCA2 can help override this DNA damage checkpoint to enhance the transition to mitosis, regardless of whether or not DNA damage has been successfully repaired [127]. This represents another mechanism, where elevated PLK1 activity has an essential function in tumor cells and could provide a targetable vulnerability.

While cells in S and G2 phases look to HR for DSB repair, those in G1 and M phases may depend on the more error prone NHEJ repair network for resolution. Unlike HR, NHEJ does not require sister chromatids to serve as template DNA and can therefore, be called upon at any point during the cell cycle. Interestingly, PLK1 has been shown to inhibit double strand DNA break repair through NHEJ during mitosis as a mechanism to prevent telomeric fusions and genomic instability [128]. PLK1 does this jointly with CDK1 through phosphorylation of 53BP1 and X-ray cross complementing (XRCC4) to inhibit their localization to DNA, thereby inhibiting NHEJ [128,129,130]. XRCC4 is a critical partner of DNA ligase IV in the completion of NHEJ [128]. PLK1-mediated phosphorylation of 53BP1 prevents binding of ubiquitinated histones and localization to DNA damage foci for signaling [129]. While PLK1 prevents erroneous telomeric fusions in mitosis, dysregulation of PLK1 could also disturb necessary double strand DNA break repairs at other stages of the cell cycle.

### 5.3. PLK1 and Telomerase

One unique DNA structure that is important to the linear chromatin maintenance is the telomere. Telomeres are repetitive DNA sequences that protect the chromosome ends from replicative degradation and erroneous recognition and ligation as DSBs [131]. Telomeres work in concert with the shelterin complex which is composed of telomeric repeat binding factor 1 and 2 (TERF1, TERF2), protection of telomeres protein 1 (POT1), TERF1 interacting nuclear factor 2 (TINF2), tripeptidyl-peptidase 1 (TPP1), and repressor activator protein 1 (RAP1) [132]. Chromosome ends are prone to dysfunction and degradation leading to CIN, particularly when telomerase action is suboptimal relative to the cell division activity [133]. When chromosome ends go uncapped by the shelterin complex, they are prone to fusion through recognition by the double strand DNA damage repair pathways, such as NHEJ, and these fusions can drastically alter both chromatin structure and sequence [133]. Human telomerase reverse transcriptase (hTERT) replicates and lengthens the telomere sequence on chromosome ends to prevent telomere shortening, which leads to cellular senescence and apoptosis. Adult human tissues do not continually express hTERT, with low or limited activity being observed at each cell division, which effectively limits the number of the allowed cell divisions. This mechanism, however, is bypassed in cancer, as activating mutations of the promoter region in the hTERT gene is one of the most frequently observed alteration contributing to oncogenesis [134,135].

In regards to this, PLK1 interacts with hTERT, and hTERT activity positively correlates with PLK1 activity, suggesting enhanced PLK1 action leads to increased telomere maintenance [131]. PLK1 inhibition results in increased telomeric fusions, suggesting a loss of the protective telomere function [136]. In this context, PLK1 could act through the stabilization of hTERT by inhibiting its ubiquitination-mediated degradation via PLK1-provided phosphorylation [131]. PLK1 also indirectly regulates hTERT activity by the stabilizing phosphorylation of tankyrase (TNKS), a poly-ADP-ribose polymerase (PARP) that enhances hTERT activity [136]. Phosphorylation of TNKS by PLK1 inhibits binding of a negative regulator of telomere length, TERF1, to telomeric DNA, allowing telomerase elongation [137]. Conversely, PLK1 also phosphorylates TERF1 and increases its affinity for the telomeres [138]. PLK1 also regulates TERF1 via PINX1 [139]. PLK1 overexpression promotes PINX1 proteasomal degradation through phosphorylation of PINX1, allowing telomerase access to the telomeres [140]. Telomere lengthening activity is often observed in cancer cells as a mechanism used to limit DNA damage caused by increased proliferation [133]. Telomere lengthening activity is in part upregulated by PLK1, providing another potential PLK1-induced vulnerability.

Telomerase inhibition in anticancer therapies has been widely pursued and trialed, as it is aimed for a target that, similarly to PLK1, is distinguished in cancer cells from normal cells almost ubiquitously [141]. However, in vitro results have been largely more promising than in vivo, with little success observed at clinical stages [142,143]. The arising issues with telomerase inhibitors have varied from induction of high levels of genomic instability in surviving cells, which actually increases tumor aggressiveness, to resistance provided by alternative telomere lengthening mechanisms [144]. Nonetheless, some drugs proved to be more successful. Thus, Imetelstat (GRN163L), a telomerase inhibitor, is about to enter Phase III clinical trials for hematologic myeloid malignancies, having previously demonstrated effectiveness in esophageal squamous cell carcinoma [145], lung cancer [146,147], and blood cancers [148,149,150,151,152] by decreasing tumor growth and increasing tumor sensitivity to radiation. A detailed dissection of the interplay between PLK1 and telomerase could provide an effective strategy for further improvements in this direction.

## 6. Targeting Cancer Cells through PLK1

PLK1 itself is an attractive target for the development of cancer therapeutics because there is a mass of evidence indicating that it is highly overexpressed selectively in tumor cells compared to the healthy adult tissues. This pattern of increased expression is reflective of a common trend of the amplification of proto-oncogenes [153]. Studies revealing PLK1 overexpression have been done in melanoma [154], non-small cell lung cancer [155], head and neck squamous cell carcinoma [156], ovarian and endometrial cancers [157,158], esophageal squamous cell carcinoma [159], hepatocellular carcinoma [160,161], hepatoblastoma [162], colorectal cancers [163], bladder cancers [2], gastric carcinoma [164], pancreatic cancer and cancer cell lines [165], gliomas [166], breast cancers [167], including triple-negative breast cancer [168], and castration-resistant prostate cancer cell lines [169]. In many cases, overexpression of PLK1 correlates with poor prognosis [155,156], tumorigenicity and aggressiveness [170], and tumor-initiating cell (TIC) propagation [171,172]. This consistency across tumor types combined with minimal expression in normal tissues and its extensive role in chromosomal instability highlights strong PLK1 potential as a target for a tumor-agnostic cancer therapy.

### 6.1. PLK1 Inhibitors as Therapeutic Agents and Associated Challenges

The combination of PLK1s role in CIN and its widespread overexpression across cancer types has led to the development and intensive study of PLK1 inhibitors as possible cancer therapeutics. The first assessment of inhibiting PLK1 as an anti-cancer strategy was done with anti-sense oligonucleotides in 2002 [173]. The first chemical inhibitors gained attention for their anti-cancer effects shortly after [174], albeit the first PLK1 inhibitors were of a broader specificity. Later, more specific inhibitors have been developed, but overall clinical outcomes are still being modest at best. This is despite many pre-clinical studies showing PLK1 inhibition reduces tumor growth in various cell line-based models [165,175,176] and accumulated evidence, showing that PLK1 inhibition can cause lethality in TICs [171,177,178]. Since PLK1 is a part of a very intricate cell cycle signaling network, the type and timing of inhibition may be one of the factors contributing to poor or inconsistent clinical results. Indeed, one study found PLK1 inhibition to cause both radiosensitization and radioresistance, depending on the timing of treatment [179], and another work found inhibition causing either metaphase arrest or G2 phase arrest depending on inhibitor doses applied [180].

ON 01910.Na is a non-competitive multi-kinase inhibitor and was one of the first PLK1 inhibitors introduced [174]. This compound also targets other kinases, including PI3K. ON 01910.Na was tested in clinical trials up to phase III in combination with gemcitabine, but the outcome of the combination therapy was found to be 19% of patients with partial response, in comparison with 13% of patients with partial response for gemcitabine alone, with an overall increase of median survival of 3 months [181]. This poor improvement in patients, along with other studies reporting relatively low specific activity of ON 01910.Na towards PLK1 compared to newer agents, such as the ATP-binding competitor BI 2536 [182] has led to this compound being considered mostly unsuccessful.

A large group of other PLK1 inhibitors are the ATP-binding competitors based on the dihydropteridinone class of compounds, which act to inhibit PLK1 kinase activity. The most well-known members of this group are BI 6727 (or volasertib) and BI 2536. BI 2536 showed a low response rate in clinical trials [183], but is still widely used in pre-clinical work for studying the effects of PLK1 inhibition in cell line models [123,169,171,182]. Volasertib is based on the same prototype as BI 2536 but has been more successful. It shows the most promise as an anti-cancer agent in acute myeloid leukemia (AML) and it was granted the FDA Breakthrough Therapy status in 2013 in order to expedite the clinical trial process in critical illnesses [184]. However, it is yet to be FDA-approved. Of the Phase II volasertib trials that have been completed, one study showed a modest increase of 5.6 versus 2.3 months with cytarabine combination therapy in AML, one study showed insufficient improvement in metastatic urothelial cancer [185], and another study showed insufficient improvement in non-small cell lung cancer [186].

A group of thymoquinone derivative inhibitors, poloxin and poloxin-2, target the PBD motif of PLK1 [187,188]. These compounds disrupt PLK1 localization and show a phenotype of chromosome segregation defects and activation of the SAC [187,189], but these compounds are still early in development. TAK960, a pyrimidodiazepinone-based PLK1 inhibitor, acts as an ATP-competing reagent and is also in early developmental stage [190,191]. NMS-P937 is yet another new generation ATP-competitive PLK1 inhibitor, identified using structure-driven drug design on the pyrazoloquinazoline scaffold [192,193]. GSK461364, an older ATP-competing inhibitor, has entered phase I clinical trials for safety. GSK461364 is less potent towards PLK2 and PLK3 over PLK1 by 300 fold [180], but was found to cause a high incidence of side-effects such as venous thrombotic emboli [194]. Most recently, highly selective ATP competitive inhibitors, PCM-075 (onvansertib) and CYC140, entered into early clinical trials (clinicaltrials.gov identifiers NCT03884829 and NCT03414034). Overall, PLK1 inhibition has not had success as a targeted cancer therapy despite high anticipation arising from promising pre-clinical investigations. This is somewhat surprising, since the rationale for targeting PLK1 is strongly supported in theory by the accumulated preclinical data and our understanding of its role in cancer cells. The lack of response to PLK1 inhibitors could be explained through either non-specific effects of the PLK1 inhibitors, or to resistance to PLK1 inhibition. In particular, adaptive resistance is displayed by CIN tumor cells [195]. For example, cancer cells adapt to decreased microtubule-kinetochore dynamics and suppressed CIN induced by treatment with KIF2C/MCAK inhibitors by adjusting AURKB levels, ultimately returning the cancer cells to their original level microtubule-kinetochore dynamics and chromosome mis-segregation [195]. As a regulator of CIN and microtubule-kinetochore dynamics, it is plausible PLK1 inhibition could elicit similar acquired resistance.

The structural similarity of the kinase and PBD domains in other PLK family members and non-specific targeting may be another source of challenges in PLK1-based therapies. The other PLK family proteins have diverse cellular roles [29]. PLK2 has a biological role in G1 phase progression and centriole duplication [196,197]. It is a p53 target gene and might function as a tumor suppressor as it is often downregulated in cancer [198,199,200]. Of interest in regard to toxicities resulting from non-specific PLK inhibition, PLK2 has a well-established role in neural plasticity and in this context, inhibitors targeting PLK2 may do more harm than help [201,202]. PLK3 has a role in Golgi fragmentation [203]. It is also suggested as a tumor suppressor due to its upregulation following p53-mediated DNA damage response and oxidative stress response, and its downregulation in some malignancies [204,205,206]. It has also been shown to promote tumor initiation in vivo [207]. PLK4 and PLK5 are structurally more divergent than PLKs 1, 2, and 3 [208] and therefore, should be to a lesser extent non-specifically affected by PLK1 inhibitors. Overall, the functions of the other PLK family proteins are much less studied in comparison to PLK1, they may have currently unknown functions critical to healthy cell biology and thus, caution is required, when using PLK inhibitors in treatment protocols [29]. This is further confounded by issues associated with the diverse functions of PLK1 itself associated with its complex contribution to CIN.

Targeting CIN to address the source of tumor heterogeneity is an appealing idea but must be approached carefully. While CIN can promote tumorigenesis in some cases such as when the increased rate of mutations gives rise to a growth advantage, it also can inhibit tumorigenesis in other cases where CIN-driven aneuploidy causes a growth defect or is incompatible with cell division [209]. The paradoxical aspect of CIN and its role in tumorigenesis is an emerging area of study in cancer. Most studies attribute this to very complex regulatory mechanisms dedicated to ensuring genomic stability, where too big or too small shift in either direction might produce undesirable effects. On the tumorigenic side, elevated CIN leads to an increased array of mutations and phenotypes in the heterogenous tumor cell population, some of which will likely confer a growth advantage or drug resistance in the altered tumor microenvironment [210]. However within the tumor suppressive response, extremely high levels of CIN are just not compatible with cell viability, as eventually, the accumulation of DNA defects will render cancer cells unable to complete mitosis or carry out other vital cell processes, causing their elimination [210]. Consistent with these ideas, it has been shown that CIN levels dictate whether it acts in a tumor suppressive or pro-oncogenic manner [209,211], and the worst prognoses are associated with low to intermediate CIN rather than its higher levels [212]. One explanation for this paradoxical behavior, CIN itself can impede tumor growth due to the increased defect burden, and so reducing this CIN burden contributes to increased cell growth, This could explain inconsistent results regarding the effect of PLK1 on tumor initiation versus tumor inhibition, with some studies showing PLK1 driving oncogenic transformation and increased growth advantage but others showing PLK1 overexpression decreasing tumor initiation rates [107,213,214,215]. Because the level of CIN determines its impact in cancer, inhibiting high CIN to intermediate tumorigenic levels is a real risk and altering PLK1 activity could produce unpredictable outcomes. Indeed, PLK1 haplo-insufficient mice displayed widespread aneuploidy accompanied by a higher incidence of tumor formation [39], and inhibition of PLK1 in an adenomatous polyposis coli protein (APC)-truncated context impaired SAC function and led to increased CIN and tumor formation [214], reminiscent of a tumor suppressor-like function in tumorigenesis. However, it should be noted that results generated in in vitro cell line work may not accurately reflect the tumor environment and selective pressures exerted on CIN cells that exist in animal models and human cancers. Inhibition of PLK1 might increase rather than decrease tumor development in some cases. These mechanisms of this context-dependent tumorigenesis versus tumor suppression may not be unique to PLK1, and may be a result of CIN regulatory feedback loops that aid cellular response to insult [195], but nonetheless, these conflicting results show the importance of the complex role of CIN and PLK1 and using a less conventional approaches to indirect targeting PLK1 overexpression in cancer. One of these strategies could be based on targeting genetic interactions of PLK1 instead of the direct inhibition of its activity.

### 6.2. Genetic Interactions of PLK1 as Therapeutic Targets

Genetic interactions represent a concept pioneered in yeast studies and have become popular in cancer research with the advent of large-scale genomic studies. Genomics and associated technologies have allowed complex genetic interaction networks to be more easily identified in comprehensive unbiased ways [216,217]. Genetic interactions are defined as an unexpected phenotype resulting from the combination of mutations whose individual mutation phenotypes are not additive, but rather synergistic [218]. These genetic interactions are grouped into two main classes, negative and positive. Negative genetic interactions result in synergistic decreases in cell viability or cell death, whereas positive genetic interactions provide a growth advantage to cells [219]. Negative genetic interactions are the focus in target discovery in cancer research, since cancer is accompanied by array of deleterious genetic alterations that provide a route for identifying a matching synergistic negative interaction to provide valuable potential targets revealed by high through-put screening. These screens allow to generate comprehensive gene interaction maps for multiple cancer-related mutations [220,221,222]. The two main types of negative genetic interactions are synthetic lethality (SL) and synthetic dosage lethality (SDL). The main difference between these two interactions is the nature of gene alterations involved. SL refers to the combination of two loss-of-function alterations, and SDL refers to the combination of a gain-of-function alteration and a loss-of-function alteration [220,221]. In case of PLK1, the aim should be to identify its SDL partners loss-of-function or inhibition of which results in cell lethality only when PLK1 overexpression is present.

It is important to reliably identify effective therapeutic targets to improve treatments, but often, as in the case of PLK1, directly targeting a seemingly promising cancer-related molecule produces strong harmful side effects or unpredictable outcomes. That is where the SDL strategy comes into play, as it provides an alternative approach to targeting overexpressed or overactivated molecules that does not rely on their direct inhibition. Instead, this method aims to inhibit their SDL partners, which can be proteins with less varied functions, which is expected to widen the therapeutic window for safe successful treatment. In particular, in the case of PLK1 and PLK1-related CIN, where too much and too little activity are both highly detrimental to cells, using an SDL approach should circumvent any risks associated with the induction of oncogenic CIN levels.

Studies proposing SL or SDL strategies as a way to indirectly target CIN seem to show more promise than approaches to directly target CIN-controlling molecules. An example of the success of this strategy is targeting the SL interaction between PARP and the breast cancer susceptibility gene (BRCA) in cancer [223,224]. PARP and BRCA both have roles in DNA damage repair, PARP through the resolution of single strand breaks and BRCA through homologous repair, and therefore they both individually control CIN. In addition to displaying SL with each other, PARP and BRCA also show SL with other targets within CIN pathways [210]. Emerging evidence has extended the list of SL of PARP to include a variety of DNA damage genes, such as MRE11 [225], ATR [226], ATM [227], and RAD54B [228]. It might be also possible to map the SL interactions between the DNA damage pathways rather than the individual genes and use this information for treatment development [210]. As shown in Figure 1A, many mitotic kinases in addition to PLK1 are elevated across different cancer types. Regardless of whether this overexpression of components regulating CIN is a mechanism of cancer cells to facilitate tumorigenicity or is a result of increased CIN and the presence of an increased number of chromosomes, as often seen in cancers, is a question for future studies, these mitotic kinases represent a source of cancer-specific alterations for SDL target study. The identification of SDL interactions of PLK1 in the processes promoting CIN, as discussed in this review, is expected to provide novel options for developing effective anti-cancer therapies that would bypass dangers associated with its direct inhibition.

## 7. Conclusions

PLK1 is a mitotic kinase near-ubiquitously overexpressed in cancer, with a key multifaceted role in the maintenance of chromosomal stability. It is also a highly attractive therapeutic target for cancer based on these characteristics and a mass of pre-clinical studies. However, PLK1 action is extremely complex and varied, and is far from being sufficiently understood. PLK1 can contribute to CIN and therefore tumor heterogeneity through the dysregulation of mitotic entry by overriding mitotic checkpoints to drive cancer cells to mitosis before DNA replication has been properly resolved or DNA damage repaired, through dysregulation of centrosome duplication, maturation, and the formation of the mitotic spindle, through premature chromosome segregation by overriding the SAC, through improper chromosome segregation due to altered kinetochore dynamics, and through altered telomere function. Using PLK1 as a therapeutic target needs to be approached expertly to avoid undesirable outcomes due to the complex role of the level of CIN and the role of PLK1 in essential cell processes. Identifying the SDL interactions of PLK1 will provide a valuable information for developing effective precision therapies for treating PLK1-overexpressing malignancies.

## Figures and Tables

**Figure 1 cancers-12-02953-f001:**
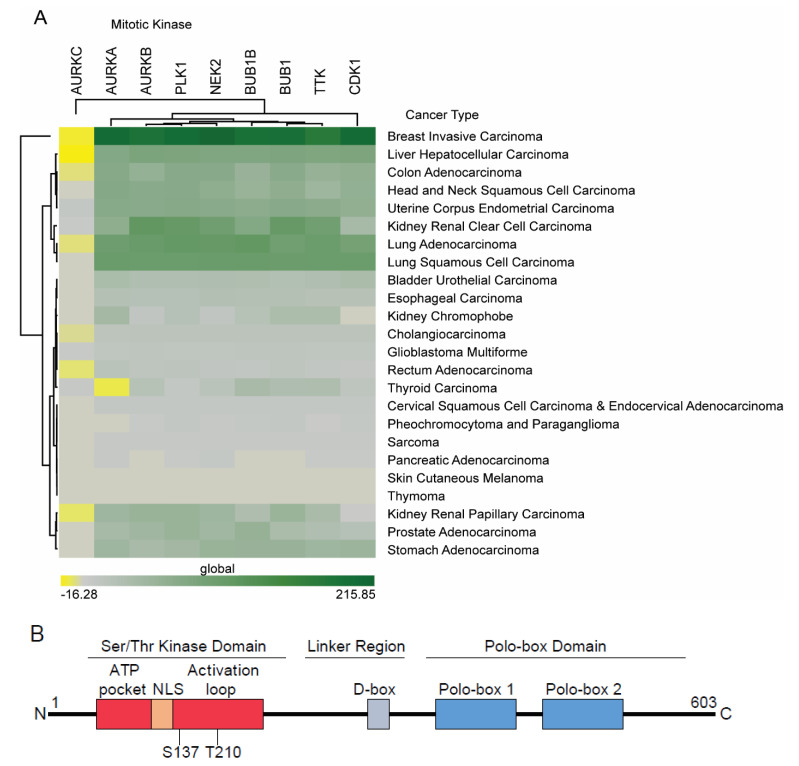
Mitotic kinases that are involved in cancer. (**A**). Differential expression of mitotic kinases across cancer types as analyzed from TCGA data. Green color indicates higher expression in cancer. (**B**). A schematic representation of PLK1 structure. This includes the Ser/Thr kinase domain, the linker region, and PBD. Activating phosphorylation at residues T210 and S137 is required catalytic activity. The D-box is the site for signaling for proteasome degradation. NLS targets PLK1 into the nucleus, while PBD targets PLK1 to the centrosome, substrates and functional partners.

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
