# Peer review of "The CINs of Polo-Like Kinase 1 in Cancer"

_cancers, 2020, doi:10.3390/cancers12102953_

Round 1
Reviewer 1 Report
In this review article, the authors focus on Plk1, a major mitotic kinase involved in the regulation of a number of processes such as centrosome function, SAC activity, kinetochore-microtubule attachments and dynamics, mitotic entry in the presence of DNA damage and in the regulation of telomere function. The authors provide an in-depth analysis of Plk1-interacting proteins during these processes and during the different stages of the cell cycle. Finally, the limitations of Plk1 inhibitors in clinical trials and the strategy of identifying synthetic dosage lethality partners of Plk1 as a promising strategy, are also explored here.
In principle, this review will be of general interest to the scientists interested in cancer cell biology, more specifically in the areas of CIN, mitosis and mitotic signaling pathways. Thus, it will be appropriate and valuable to the readership of the journal Cancers. Overall, this review is well written and clear, but there are a few points that require more attention, thus precluding me from recommending this for publication in its current format. I've outlined some of these issues below to help the authors craft a more impactful review on this subject matter.
Main points
- At some points, the text is not written in a manner that makes the material easily accessible and doesn't do a very good job of summarizing a synthetic view of the concepts presented. I would suggest revisiting the more descriptive chapters (chapters 3-5) taking this into account.
- The review fails to provide a unified message. In attempting to cover all aspects of Plk1 interacting partners and pathways, no clear message shines through. I would recommend that the authors focus more on the clinical outcomes of Plk1 inhibition and explore other therapeutic avenues for the targeting of Plk1 in cancer. The authors focus on the results of near-ubiquitous Plk1 deregulation in cancer cells but the section on the therapeutic approaches that show most promise should be developed for a more scholarly and interesting review article.
- The complex relationship between the regulation of Plk1 and Aurora B has been thoroughly documented by several labs and this should be acknowledged in this review. This feedback regulation between Plk1 and Aurora B signaling may aid in the interpretation of the clinical trial data with Plk1. And in agreement, Aurora B inhibitors have also failed to triumph in clinical trials.
- The data in Fig1A show that all of the kinases analyzed are mostly overexpressed in several tumor types, suggesting that Plk1 is not an exclusive target in this regard. The data are also consistent with a general overexpression of proteins involved in controlling chromosome segregation, which could be explained by the larger number of chromosomes in CIN cancer cells. How do we know that Plk1 overexpression is an advantage for CIN cancer cells rather than a consequence of having more chromosomes? The data also support a model in which a more unspecific kinase inhibitor may help to target more of the essential kinase pathways. However, the unspecific effects observed after Plk1 inhibition alone suggest that across-the-board kinase inhibition in cancer cells may generate even more adverse effects. Can the authors explore this in the discussion?
Minor points
- Line 45 “Normally these checkpoints ensure...” – this sentence is confusing and should be revised
- Line 64 - Roles in chromosome condensation for Aurora B (via pH3) and roles in centrosome clustering and MT nucleation at the midzone for Aurora A have also been described and these are important for the regulation of CIN
- Line 91 – Should read “The PBD”
- Line 102 - Should read "PLK1 protein levels peak..."
- Lines 131-133 - Sentence is long and confusing
- Line 144 - Maybe uni-directional instead of irreversible. Possibly change this sentence all together since the feedback loops themselves are not really thought of as being irreversible. What is irreversible, is the transition to different stages of the cell cycle.
- Line 151 - Missing a comma after PLK1 itself
- Line 175 - Should read "In this context, CIN can arise from..."
- Lines 181-182 - The data so far have shown that Plk1 depletion prevents centrosome duplication, but whether overexpression of Plk1 results in supernumerary centrosomes is still a matter of debate. To my knowledge, this has only been shown in an LKB1-depleted background, so this would probably be represented in a small subset of CIN tumors at best. However, an essential role for Plk4 in centrosome duplication has been amply documented. Nevertheless, it appears clear that Plk1 activity is important for centrosome function and I would suggest the authors to revise the text accordingly.
- Lines 202-204 - The relationship between Plk1 and AurA should also be included here
- Line 205 - Should read "cancer cells to form"
- Line 224 - Should read "condensin complex"
- Line 239 - Should read "SAC function result"
- Title of section 4.3 Should read "PLK1 and kinetochore-microtubule dynaimcs"
- Line 256 - Do the authors mean mis-segregation?
- Line 260 - The SAC does not phosphorylate proteins. It is a checkpoint and some SAC components do phosphorylate proteins. This sentence should be revised accordingly.
- Line 268 - The authors should refer to "the SAC" and not just "SAC" throughout the text
- Line 271 - Should read "segregate during anaphase"
- Line 277 - MCAK itself has been shown to be directly involved in the tuning of CIN levels in cancer cells and this is another route directly shown to lead to CIN
- Line 382-384 - Sentence is long and confusing
- Lines 459-463 - One alternative hypothesis is that cancer cells can display adaptive resistance, particularly to those compounds that are actively involved in the homeostatic control of feedback loops. And Plk1 is a prime candidate to do so. For example, previous reports have demonstrated that cancer cells can adapt to an inhibitor of CIN by tuning Aurora B levels. What happens to Plk1 levels in tumors after treatment with the Plk1 inhibitors? Are these maintained at the same levels? Cellular adaptation, whether it be through adaptive or acquired resistance is of considerable importance for clinical outcome and may indeed be one of the most notable barriers for cancer treatment. The authors should include this somewhere in the text
- Lines 486-487 - The mechanisms that govern cancer cell death have been puzzling scientists for decades and several hypotheses have been proposed (e.g. immune system clearance, loss of cellular attachment, amino acid starvation, necrosis, loss of growth factor supply, etc) and therefore it would be naive to presume that high CIN levels will only cause cell death through mitotic failure after accumulation of DNA damage. The authors should revisit this sentence and make appropriate amendments.
- Lines 497-500 - The authors should consider the alternative hypotheses:
- The fact that Plk1 inhibition paradoxically confers an advantage to tumor cells and increases tumor formation may not be specific to Plk1, but rather to the mechanisms regulating CIN, which rely heavily on feedback control. The dynamic nature of these feedback loops is precisely what confers cancer cell robustness in the presence of cellular insults, such as Plk1 inhibition or inhibition of any other major mitotic kinase such as Aurora B.
- Cellular adaptation to the manipulation of CIN may be more efficient in animal systems and therefore the in vitro cellular work often does not translate accordingly to clinical trial outcome.
- Thirdly, it has been shown that CIN imposes a burden on tumor cell growth and that reducing the burden by lowering CIN may paradoxically result in increased levels of tumor formation. However, whether these tumors are indeed more heterogenenous and whether they are more treatable, remains unknown.
- Line 524 - The other advantage of inhibition of SDL partners is the potential to target a protein with fewer cellular functions and interactors, thus reducing the probability of unpredicted side-effects.
- Line 543 - Sentence is confusing, please revisit.
- Line 552 - It is a reductionist view to attribute all undesirable effects due to changes in CIN levels and the authors should adjust their wording accordingly. Indeed, the accurate measurement of CIN in vivo is still a daunting task and largely unreliable due to the high intra-tumor heterogeneity conferred by CIN itself.
Author Response
Thank you for reviewing our work. We have now made modifications to the manuscript, and please see the attachment for these changed outlined and addressed point-by-point.
We hope that these changes have addressed the Reviewer’s concerns. Thank you once again.

Reviewer 2 Report
The manuscript is well written and all the relevent information is presented regarding the potenial use of targeting PLK interaction partner for better treatement outcomes. The author need to do minor revision including editing the manuscript.
Line 217: Dysregulation of condensin II through PLK1 inhibition leads to segregation defects such as lagging chromosomes (79). The author needs to be specific regarding the lagging chromosome. Is it telephase specific or anaphase specific chromosome lagging?
Line 267 -268 “Interestingly, phosphorylation of PLK1 at serine 137 (S137) has been shown to bypass the constraints of SAC (45,91). This would further promote CIN in the context of PLK1 overexpression. This statement a bit confusing and the author may simplify.
Line 346-47 the author clam that “PLK1 has been shown to inhibit double strand DNA break repair during mitosis as a mechanism to prevent telomeric fusions. Is it homologous recombination repair or non-homologous end joining or both? DSBs repair via classical NHEJ pathway is a predominant repair pathway to repair DSBs in mitosis (Lieber MR, Plos Genetics (2010).
Line 478 “While CIN can promote tumorigenesis in some cases, it also can inhibit tumorigenesis in others.” Can the author explain how PLK induced CIN prevent tumorgenesis?
Line 334-340 the author need to rewrite the following statement clearly. Is it the PLK1 mediated phosphorylation stimulate RAD51 dependent HR pathways? Is BRCA2 phosphrylation by PLK 1 enhances mitotic progression rather than DSB repair? “PLK1 directly phosphorylates the Rad51 recombinase, stimulating a cascade of events that facilitate further phosphorylation by CHEK2 and subsequent binding to the MRN complex, via Nbs1 (123). This response not only increases the presence of Rad51 at sites of DNA damage, but it also enhances the overall responsiveness of the HR pathway, contributing to cell proliferation and tumorigenesis. PLK1 also phosphorylates BRCA2, a stabilizer of Rad51 that promotes HR and when BRCA2 is phosphorylated, it enhances mitotic progression, regardless of whether or not DNA damage has 340 been successfully repaired (124).”
Author Response

(The authors gave the same response as above.)

Round 2
Reviewer 1 Report
The authors have successfully addressed all concerns raised during the revision process and I recommend this manuscript for publication in its current format. However, I do feel the simple abstract (added in this new version) deserves a further revision to ensure clarity .
The simple summary should be written in layman's terms and make the overall message accessible to a broader readership.
Specifically
- I suggest the use of "targeted therapies", rather than "cancer therapeutics" (lines 14 and 20) which is not the same.
- Plk1 is not an alteration specific to cancer
- The final sentence of the summary is also confusing and should be revised
- Minor improvements to the overall clarity and quality of the text would be appreciated
Author Response
Dear Editor and Reviewer,
We have now modified the simple abstract as pasted below.
Kindly let us know if you need any further details.
Thank you,
Franco Vizeacoumar
Simple Abstract: Many alterations specific to cancer cells have been investigated as targets for targeted therapies. Chromosomal instability is a characteristic of nearly all cancers that can limit response to targeted therapies by ensuring the tumor population is not genetically homogenous. Polo-like Kinase 1 (PLK1) is often up regulated in cancers and it regulates chromosomal instability extensively. PLK1 has been the subject of much pre-clinical and clinical studies, but thus far, PLK1 inhibitors have not shown significant improvement in cancer patients. We discuss the numerous roles and interactions of PLK1 in regulating chromosomal instability, and how these may provide an avenue for identifying targets for targeted therapies. As selective inhibitors of PLK1 showed limited clinical success, we also highlight how genetic interactions of PLK1 may be exploited to tackle these challenges.